# Developing health and environmental warning messages about red meat: An online experiment

Lindsey Smith Taillie[1,2]*, Carmen E. Prestemon[1], Marissa G. Hall[1,3,4], Anna H. Grummon[5,6], Annamaria Vesely[2], Lindsay M. Jaacks[7]

1 Carolina Population Center, University of North Carolina, Chapel Hill, NC, United States of America, 2 Department of Nutrition, University of North Carolina Gillings School of Global Public Health, Chapel Hill, NC, United States of America, 3 Department of Health Behavior, University of North Carolina Gillings School of Global Public Health, Chapel Hill, NC, United States of America, 4 Lineberger Comprehensive Cancer Center, University of North Carolina, Chapel Hill, NC, United States of America, 5 Center for Population and Development Studies, Harvard TH Chan School of Public Health, Cambridge, MA, United States of America, 6 Department of Population Medicine, Harvard Medical School and Harvard Pilgrim Health Care Institute, Boston, MA, United States of America, 7 Global Academy of Agriculture and Food Security, The University of Edinburgh, Edinburgh, United Kingdom

* taillie@unc.edu

**Data Availability Statement:** All data files are available from the Harvard Dataverse database (https://doi.org/10.7910/DVN/KE1KLH).

## Abstract

### Introduction

The United States has among the highest per capita red meat consumption in the world. Reducing red meat consumption is crucial for minimizing the environmental impact of diets and improving health outcomes. Warning messages are effective for reducing purchases of products like sugary beverages but have not been developed for red meat. This study developed health and environmental warning messages about red meat and explored participants' reactions to these messages.

### Methods

A national convenience sample of US red meat consumers ($n = 1,199$; mean age 45 years) completed an online survey in 2020 for this exploratory study. Participants were randomized to view a series of either health or environmental warning messages (between-subjects factor) about the risks associated with eating red meat. Messages were presented in random order (within-subjects factor; 8 health messages or 10 environmental messages). Participants rated each warning message on a validated 3-item scale measuring perceived message effectiveness (PME), ranging from 1 (low) to 5 (high). Participants then rated their intentions to reduce their red meat consumption in the next 7 days.

### Results

Health warning messages elicited higher PME ratings than environmental messages (mean 2.66 vs. 2.26, $p<0.001$). Health warning messages also led to stronger intentions to reduce red meat consumption compared to environmental messages (mean 2.45 vs. 2.19,

**Funding:** This work was funded through a grant from the Wellcome Trust, grant ID #216042/Z/19/Z. Grant #K01HL147713 from the National Heart, Lung, and Blood Institute of the NIH supported MGH's time on the paper. We are grateful to the Carolina Population Center and its NIH Center grant (P2C HD050924) for general support. The funders had no role in study design, data collection and analysis, decision to publish, or preparation of the manuscript.

**Competing interests:** The authors have declared that no competing interests exist.

$p$<0.001). Within category (health and environmental), most pairwise comparisons of harms were not statistically significant.

## Conclusions

Health warning messages were perceived to be more effective than environmental warning messages. Future studies should measure the impact of these messages on behavioral outcomes.

## Introduction

Per capita red meat consumption in the United States is among the highest in the world. On average, adults in the US consume 284 grams of unprocessed red meat per week [1] (approximately 2.5-quarter pound hamburger patties), nearly three times the recommended level identified in the landmark 2019 EAT-*Lancet* report *Healthy Diets from Sustainable Food Systems* [2]. Unprocessed red meat (defined as meat from mammals, e.g., beef, pork, lamb, goat) contains some beneficial nutrients, including protein and micronutrients such as iron and vitamin $B_{12}$ [3]. However, in the US, the vast majority of the population consumes adequate protein [4] and vitamin $B_{12}$ [5]. Additionally, while iron deficiency anemia is the most common nutrient deficiency in the US, prevalence is relatively low at 10.4% for females and 5.2% for males [6]. At the same time, high intake of red meat consumption is associated with a number of poor health outcomes, including increased risk of several types of cancer, type 2 diabetes, high blood pressure, high cholesterol, and premature death [7,8]. These relationships are particularly strong for processed red meats that are salted, cured, fermented, smoked or otherwise processed for flavor or preservation [9]. According to the American Cancer Society, it is unknown if there is a safe level of red meat consumption [10]. In lieu of knowledge of safe consumption levels, organizations like the American Cancer Society and the American Diabetes Association recommend limiting red meat consumption in favor of poultry, fish, or plant-based proteins [10,11].

Red meat production is also damaging to the environment. Livestock production is responsible for 14.5% of anthropogenic greenhouse gas (GHG) emissions globally [12]. Among animal products, red meat is particularly damaging; for example, beef contributes approximately five times as much GHG emissions compared to chicken [13]. Red meat production also damages the environment through deforestation and excess water use [14]. International reports by experts on food systems and climate change, including the EAT-*Lancet* report and the Intergovernmental Panel on Climate Change (IPCC) report, indicate a drastic reduction in red meat consumption in high-income countries is needed to prevent further damage to the climate [15].

Given this evidence, public health practitioners and policymakers have sought strategies for reducing red meat consumption in the US. One potential population-level strategy is to use front-of-package labels to inform consumers about health and environmental effects of red meat. For example, several countries have mandatory or voluntary carbon labelling on food products, and given the large carbon footprint of red meat in particular, these labels may specifically discourage red meat consumption [16]. Additionally, several private companies across the globe have also implemented similar optional "green label" certification for products [17]. One online experiment found that providing consumers with information about GHG emissions of vegetarian and meat-containing soups reduced intentions to purchase the meat-

containing soups [18]. Studies of similar carbon footprint labels and behavioral nudges on meat products in cafeteria settings in Europe have also found reductions of meat-purchasing behaviors [19–21].

However, these studies focused on numerical labels about carbon emissions, rather than warning messages that make a direct statement about the health or environmental harms of a product. Previous research has highlighted the promise of warning labels to reduce purchases and consumption of tobacco products [22,23], sugary beverages [24,25], and junk food and alcohol [26]. Yet, currently there is very little knowledge about warning labels on red meat. Specifically, limited research has compared health and environmental warning messages side by side or identified the most promising health and environmental harms to include in warning messages. We recently published a study comparing two specific health warnings (about cancer and early death) and two specific environmental warnings (about environmental damage and carbon emissions) [27]; here, we describe the exploratory research that led to the development of the specific warnings used in that study.

Research is also needed to assess potential differences in responses to the warning messages based on demographic characteristics, in order to inform policies to promote equity. A systematic review on consumer perspectives towards environmental concerns of red meat consumption found that females are more likely than males to be willing to reduce meat consumption after being exposed to environmental harm messages [28]. Individuals with higher education levels tend to comprehend high-literacy sustainability and carbon labels compared to those with less education [29]. Other demographic characteristics might also affect responses to red meat warnings, but limited research has explored this question.

The objective of this study was to develop health and environmental warning messages about red meat consumption and examine US consumers' reactions to these messages, with the overarching goal of identifying which messages hold the most promise for use in future studies to test behavioral impact. We further aimed to assess potential differences in message reactions across demographic groups. Our primary outcome was perceived message effectiveness (PME), a measure that has been used extensively in similar experiments to identify the potential impact of food and beverage warning labels on consumers [27,30–34]. PME is a measure that is sensitive enough to detect small differences between warnings yet is also predictive as to messages' ability to change actual behaviors [35].

## Methods

### Message development

We designed messages using a multistage process similar to studies to evaluate health warnings for other products [31,36]. First, we conducted a review of the literature on the health and environmental outcomes associated with red meat to select outcomes. Outcomes were included if recent meta-analyses, systematic reviews, or expert consensus statements indicated an association between red meat consumption and the outcome (**Table 1**). High-quality longitudinal cohort studies are included in the below table if they were published after the most recent systematic review or meta-analysis on the topic.

Based on this literature review, we created 14 health messages and 16 environmental messages. We then convened an Expert Advisory Committee comprised of experts in nutrition, environment, food policy, agriculture policy, and law to review and provide feedback on both the warning message wording and scientific evidence supporting each of the harms. From this meeting, we created a final list of eight health messages and ten environmental messages to be used in this experiment. The number of messages selected for health and environment reflected the number of distinct topics for which we found scientific support (**Table 1**), as well

**Table 1. Health and environmental outcomes selected for warning messages and supporting literature.**

| Health Outcome | Key Citations | Professional Association Dietary Recommendation for Red Meat Consumption |
|---|---|---|
| Type 2 diabetes | [37,38] | American Diabetes Association [39]:<br>Recommendation: "limiting intake" of red meat for diabetes prevention. |
| Several types of cancer<br>Colon and rectal cancer<br>Colon cancer | [40–42] | American Cancer Society [43]:<br>Notes: "It is not known if there is a safe level of consumption for either red or processed meats."<br>Recommendation: "choosing protein foods such as fish, poultry, and beans more often than red meat, and for people who eat processed meat products to do so sparingly, if at all." |
| Cardiovascular disease<br>Heart damage | [44] | American Heart Association [45]:<br>Notes: "In general, red meats (beef, pork and lamb) have more saturated (bad) fat than chicken, fish and vegetable proteins such as beans. Saturated and *trans* fats can raise your blood cholesterol and make heart disease worse."<br>Recommendation: "Minimize processed red meats like bacon, ham, salami, sausages, hot dogs, beef jerky and deli slices. . .Choose nonfried fish, shellfish, poultry without the skin, and trimmed lean meats, no more than 5.5 ounces, cooked, per day." |
| Stroke | [46,47] | The American Stroke Association is a division of the American Heart Association; see above recommendations from the American Heart Association [45]. |
| Early death | [48,49] | N/A no professional association |
| **Environmental Outcome** | **Key Citations** | |
| Climate change/ global warming | [12,50] | |
| Climate change, which leads to extreme weather events | [51] | |
| Water pollution | [50,52] | |
| Carbon footprint/greenhouse gas emissions | [13] | |
| Deforestation | [15] | |
| Water shortages | [52,53] | |
| [harms the] environment/ planet | [2,50] | |

as discussions with the expert advisory committee. The health harms were: diabetes; several types of cancer; colon cancer; colon and rectal cancer; cardiovascular disease; heart damage; stroke; and early death (e.g., "Eating red meat increases your risk of early death"). Environmental harms were: climate change; climate change, which leads to extreme weather events; global warming; carbon footprint; greenhouse gas emissions; water pollution; water shortages; deforestation; environment; and planet. We chose to include two messages about climate change (one general and one that linked climate change to extreme weather events) to assess whether consumers have stronger reactions to a climate-related message when a concrete environmental outcome (like extreme weather) is used, as the general population often has difficulty with understanding terminology related to climate change [54]. The messages tested are presented in **Table 2**.

To create stimuli for the survey, we mimicked current nutrient warning labels on food products in Latin America with regards to color (black) and shape (octagonal) [55]. We used an octagon-shaped label and included the prefix 'WARNING' because these features have been found to be effective in attracting attention and maximizing message effectiveness among US adults [31]. Example stimuli are shown in **Fig 1**.

## Participants

From March 22-March 25, 2020, we recruited a sample of 1,199 U.S. adults aged 18 years and older using Cloud Research's Prime Panels, an online participant recruitment company commonly used by social and behavioral science researchers [56]. Eligibility criteria were being

**Table 2. Text of health and environmental warning messages tested in online experiment.**

| Health Warning Messages | Environmental Warning messages |
|---|---|
| **Eating red meat increases your risk of**... | **Eating red meat**... |
| type 2 diabetes. | contributes to climate change. |
| several types of cancer. | contributes to climate change, which leads to extreme weather events. |
| colon cancer. | contributes to global warming. |
| colon cancer and rectal cancer. | increases your carbon footprint. |
| cardiovascular disease. | increases your greenhouse gas emissions. |
| heart damage. | contributes to water shortages. |
| stroke. | increases water pollution. |
| early death. | increases deforestation. |
| | harms the environment. |
| | harms the planet. |

aged 18 years or older and consumption of any red meat in the past 30 days. We used an item adapted from the National Health and Nutrition Examination Survey (NHANES) to measure consumption of red meat in the past 30 days ("In the past 30 days, how often did you eat red meat? Red meat includes beef, lamb, pork, sausage, and ham. It also includes processed red meats such as bacon, hot dogs, and lunch meats. It does not include chicken, turkey, or seafood products." Response options: Never, Less than once a week, Once a week, 2–3 times a week, 4–6 times per week, 1 time per day, 2 times per day, 3 or more times per day) [57]. No *a priori* power calculations were conducted for this exploratory study.

## Procedures

The exploratory study was conducted online using the survey platform Qualtrics. Median completion time was 10 minutes. The main experiment employed a mixed between-within design (**Fig 2**). Eligible participants were randomized to view a series of either health or environmental warning messages (between-subjects factor). The within-subjects factor was the specific health or environmental harm included in the warning message. Participants randomized to the health arm viewed a series of eight health warning messages, and those randomized to the environmental arm viewed a series of ten environmental warning messages (**Table 2**).

**Fig 1. Example of health and environmental warning images tested in online experiment of 1,199 US adults (March 2020).**

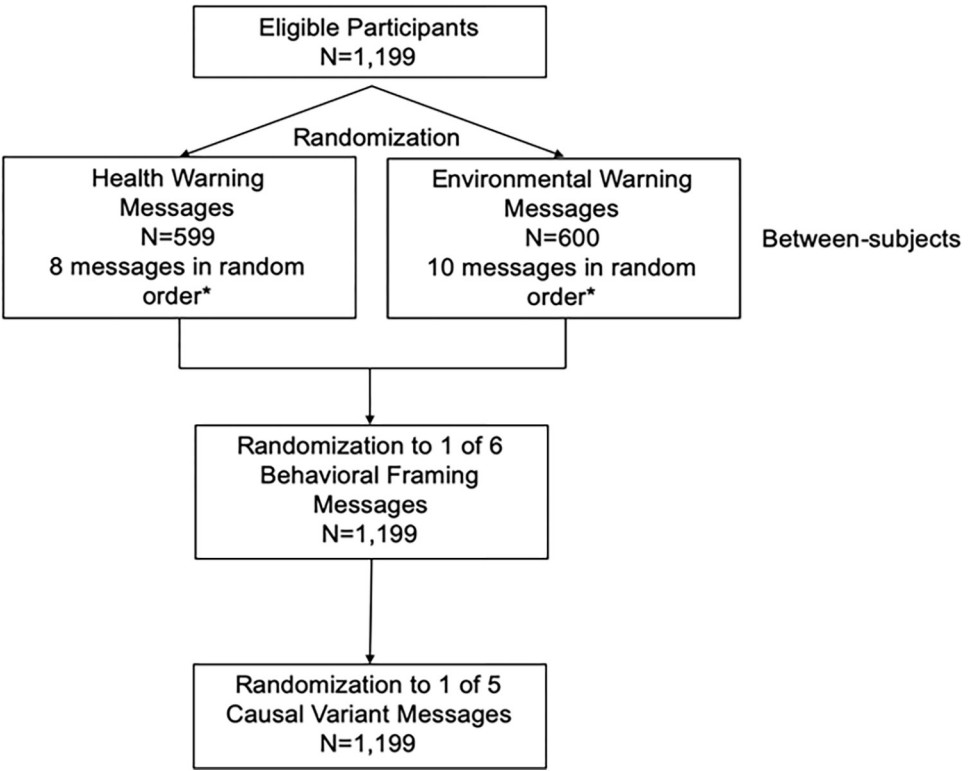

**Fig 2. Participant flow diagram of online experiment of 1,199 U.S. adults (March 2020).** *Within-subjects factor was the 8 health and 10 environmental messages.

Similar to previous message development studies [58], this study omitted a control group, because the main purpose was to understand whether health or environmental messages were perceived as more effective relative to each other.

Participants viewed specific messages in random order. After viewing each message, participants answered survey questions about the message (**S1 Appendix**). After viewing all messages, participants were asked to select which of the messages most discouraged them from wanting to eat red meat.

Then, participants completed two additional experimental tasks to examine other important but often overlooked design choices for warning messages. First, we examined consumer reactions to differences in the framing of how the target behavior was described (behavioral framing task). All participants were randomized to view one of the following six messages: 1) Eating red meat contributes to species loss; 2) Eating too much red meat contributes to species loss; 3) Eating a lot of red meat contributes to species loss; 4) Consumption of red meat contributes to species loss; 5) Overconsumption of red meat contributes to species loss; 6) Excess consumption of red meat contributes to species loss. (Underlining is presented here to highlight differences between conditions but was not used in the survey).

Second, we examined reactions to messages with varying strength in causal language (causal variant task). Prior research on causal variants has shown that 'causes' is more discouraging than weaker variants such as 'contributes to,' 'can contribute to,' and 'may contribute to' [58]. This study aimed to test additional variations of causal language that could be applied to red meat. All participants were randomized to view one of five messages: 1) Eating red meat contributes to obesity; 2) Eating red meat may contribute to obesity; 3) Eating red meat is

associated with obesity; 4) Eating red meat increases your risk of obesity; 5) Eating red meat increases your chances of obesity. (As above, underlining is presented here to highlight differences between conditions but was not used in the survey).

## Measures

The primary outcome for the main experiment was PME of the health and environmental warning messages. After viewing each message, participants rated the message on a three-item PME scale. We adapted items from the UNC PME scale, which has previously been validated for use developing tobacco warnings [59]. The three PME items used in this study were: "How much does this message… 1) discourage you from wanting to eat red meat?; 2) make eating red meat seem unpleasant to you?; 3) make you concerned about the health/environmental effects of eating red meat?" The final item regarding concern about health/environmental effects was tailored to participants' assigned condition (i.e., participants in the health warning message condition were asked about health effects and those in the environmental warning message condition were asked about environmental effects). The three PME items used five-point Likert-type response options ranging from "Not at all (coded as 1) to "A great deal" (coded as 5). Average PME (the primary outcome) was calculated by averaging responses across all three items.

Secondary outcomes for the main experiment were intentions to limit red meat consumption over the next seven days and selection of the most discouraging message in each message category (health and environmental). Intention to reduce red meat consumption in the next seven days [60] was measured using the same five-point Likert response options as PME. To measure the most discouraging message, participants were asked "Which of these messages most discourages you from wanting to eat red meat?" (underline shown in survey). The answer choices were the eight health or ten environmental warning messages (dependent upon condition).

The outcome for the behavioral framing and causal variant tasks was perceived discouragement, measured using the discouragement item of the UNC PME scale.

After completing the experiment, participants answered a series of questions on demographic characteristics as well as political affiliation and change in red meat consumption over the past 12 months.

## Ethics

The University of North Carolina Chapel Hill Institutional Review Board approved this study (protocol #: 19–3349). All participants provided online written informed consent. Participants received incentives in cash, gift cards, or reward points from Prime Panels.

## Statistical analysis

All analyses were conducted in Stata/SE version 14.1 (StataCorp LLC, College Station, TX). For all analyses, $p < 0.05$ was considered statistically significant. The study protocol and hypotheses were preregistered on Aspredicted.org, prior to data collection (https://aspredicted.org/kp28j.pdf).

We used chi-square tests to check that socio-demographic characteristics were balanced across arms (**Table 3**). Of the nine balance tests conducted, 1 was significant: participants randomly assigned to the environmental arm were slightly less likely to report consuming less meat as compared to a year ago (24.8%) relative to participants randomly assigned to the health warning arm (23.4%) ($p = 0.04$).

**Table 3. Characteristics of participants in online experiment of health and environmental warning messages for red meat (*n* = 1,199) from March 2020 in the United States.**

| Characteristic | All (n = 1,199) | Health Warnings (n = 599) | Environmental Warnings (n = 600) | p-value* |
|---|---|---|---|---|
| | N (%) | N (%) | N (%) | |
| **Age categories n (%) †** | | | | |
| 18–25 years | 200 (16.8) | 94 (15.8) | 106 (17.9) | 0.25 |
| 26–34 years | 214 (18.0) | 114 (19.2) | 100 (16.9) | |
| 35–44 years | 201 (16.9) | 106 (17.8) | 95 (16.0) | |
| 45–54 years | 183 (15.4) | 96 (16.1) | 87 (14.6) | |
| 55–65 years | 195 (16.4) | 97 (16.3) | 98 (16.5) | |
| 65 years and older | 195 (16.4) | 88 (14.8) | 107 (18.0) | |
| **Gender, n (%)** | | | | |
| Male | 592 (49.4) | 287 (47.9) | 305 (50.8) | 0.62 |
| Female | 603 (50.3) | 308 (51.4) | 295 (49.2) | |
| Non-Binary | 3 (0.2) | 3 (0.5) | 0 (0) | |
| Self-Describe | 1 (0.08) | 1 (0.2) | 0 (0) | |
| **Race, n (%) ‡** | | | | |
| White | 993 (82.3) | 498 (83.1) | 495 (82.5) | 0.54 |
| Black or African American | 131 (10.9) | 62 (10.3) | 69 (11.5) | |
| Asian | 30 (2.5) | 30 (5.0) | 30 (5.0) | |
| Pacific Islander | 2 (0.2) | 0 (0) | 2 (0.3) | |
| Native American or Alaskan Native | 31 (2.6) | 16 (2.7) | 15 (2.5) | |
| Other Race Not Listed | 24 (2.0) | 11 (1.8) | 13 (2.2) | |
| **Ethnicity, n (%) §** | | | | |
| Hispanic/Latinx | 118 (9.8) | 63 (10.3) | 55 (9.2) | 0.81 |
| Not Hispanic/Latinx | 1081 (90.1) | 536 (89.5) | 545 (90.8) | |
| **Education Level, n (%)** | | | | |
| High School Degree or Less | 475 (39.6) | 244 (40.7) | 231 (38.5) | 0.97 |
| Associate or Technical Degree | 269 (22.4) | 134 (22.4) | 135 (22.5) | |
| 4-year College Degree | 314 (26.3) | 146 (24.4) | 168 (28.0) | |
| Master's, Graduate, or Higher | 140 (11.7) | 75 (12.5) | 65 (10.8) | |
| **Household Income, annual, n (%)** | | | | |
| Less than $24,999 | 276 (23.0) | 146 (24.4) | 130 (21.7) | 0.91 |
| $25,000 to $49,999 | 328 (27.4) | 161 (20.4) | 167 (27.8) | |
| $50,000 to $74,999 | 253 (21.1) | 132 (22.0) | 121 (20.2) | |
| $75,000 to $99,999 | 154 (12.8) | 74 (12.4) | 80 (13.3) | |
| $100,000 or more | 188 (15.7) | 86 (13.9) | 102 (16.8) | |
| **Political Affiliation, n (%)** | | | | |
| Liberal | 311 (25.9) | 159 (26.5) | 152 (25.3) | 0.06 |
| Moderate | 522 (43.5) | 251 (41.9) | 271 (45.2) | |
| Conservative | 363 (30.3) | 189 (31.5) | 174 (29.0) | |
| **Frequency of Red Meat Consumption, n (%)** | | | | |
| Less than 1 per week | 109 (9.1) | 48 (8.0) | 61 (10.2) | 0.72 |
| 1 time per week | 170 (14.2) | 99 (16.5) | 71 (11.8) | |
| 2–3 times per week | 552 (46.0) | 268 (44.7) | 284 (47.3) | |
| 4–6 times per week | 238 (19.8) | 122 (20.4) | 116 (19.3) | |
| 1 time or more per day | 130 (10.8) | 62 (10.4) | 68 (11.3) | |
| **Red Meat Consumption as Compared to a Year Ago, n (%)** | | | | |
| Less | 379 (31.7) | 184 (30.8) | 195 (32.6) | 0.04 |

*(Continued)*

**Table 3.** (Continued)

| Characteristic | All (n = 1,199) | Health Warnings (n = 599) | Environmental Warnings (n = 600) | p-value[*] |
|---|---|---|---|---|
| | N (%) | N (%) | N (%) | |
| About the Same | 742 (62.0) | 374 (62.6) | 368 (61.4) | |
| More | 75 (6.3) | 39 (6.5) | 36 (6.0) | |
| **Believe that Human Activity is Main Cause of Climate Change, n (%)** [‖] | | | | |
| Strongly Disagree | – | – | 46 (7.7) | – |
| Somewhat Disagree | – | – | 60 (10.0) | |
| Neither Agree nor Disagree | – | – | 67 (11.2) | |
| Somewhat Agree | – | – | 219 (36.5) | |
| Strongly Agree | – | – | 208 (34.7) | |

[*]P-values were calculated using chi-square tests.

[†]Non-reported ages (n = 11) were set to missing.

[‡]Percents of race do not add up to 100% due to participants being able to select multiple races. There were 32 participants that selected multiple races. Due to small cell size, races other than Black were collapsed into an 'other race' category for regression analysis.

[§]Listed ethnicities: Mexican/Mexican American/Chicano, Cuban, Other Listed (Columbian, Dominican, Ecuadorian, Panamanian, Puerto Rican, Spanish, Venezuelan, Peruvian, Other).

[‖]Climate change belief questions were only shown to those that were randomized to the climate warning message condition.

To assess whether PME varied by message type (health vs. environmental) in the main experiment, we used a t-test to compare PME ratings (averaged across all warning messages) between the two conditions. Results did not differ in direction or statistical significance when we used linear regression controlling for the unbalanced covariate (change in red meat consumption) (**S1 Table**), so we retained the unadjusted analysis. We also used t-tests to evaluate differences in intention to reduce meat consumption over the next seven days.

To compare the different messages within topic in the main experiment, we used mixed-effects linear regression, with separate models for health and for environmental messages. For both models, indicator variables were used for each warning message (excluding one), treating the intercept as random to account for repeated measures. We used postestimation commands to conduct pairwise comparisons of the predicted means, correcting for multiple comparisons. The reported p-values for these comparisons were produced using Bonferroni's method.

To evaluate the results of the most discouraging message, we used the descriptive results of the frequency of selection of warning messages. Additionally, we conducted z-tests to assess statistical significance in the frequency of selection of the different messages.

In exploratory analyses, we used multivariate linear regression with OLS estimation to examine predictors of average PME for health and environmental messages (averaged across all warning messages shown in **Table 2**). Predictors were chosen based on characteristics previously found to be associated with food purchases and dietary intake overall and red meat intake in particular [61,62]. Separate regressions were conducted for health and environmental messages. The following predictors were included in the models: demographic characteristics (age, gender, race, Latinx ethnicity, income, education level, household income), political affiliation, usual red meat consumption, and change in red meat consumption in the past 12 months. For the environment model, belief in climate change was also included.

There were four participants that selected "nonbinary" or "self-describe" for their gender; these respondents were not included in the OLS regressions involving gender due to small cell size. Racial categories other than Black and white were collapsed into an "other race" category and analyzed as such in the OLS due to small sample size in other racial categories. Likewise,

those who had no change in red meat consumption over the past 12 months were combined with those who had increased their red meat consumption over the past 12 months due to the fact that only 6.3% of the sample increased their red meat consumption over the past 12 months, and a dichotomous variable of "reducers" and "non-reducers" was used for analysis. In post estimation tests, no variable had a Variance Inflation Factor of 5 or greater, indicating that multicollinearity was not problematic and thus all variables were retained in the model.

For the behavioral framing and causal variant tasks, we assessed whether PME varied by message by conducting a one-way ANOVA to examine statistical differences between groups.

## Results

Participants were, on average, 44.6 years old (range: 18–95). About half (50.3%) were female, and 82.3% were White (**Table 3**).

### Differences between health and environmental messages

Health warning messages (mean PME 2.66, SD 1.18) were perceived as more effective than environmental warning messages (mean PME 2.26, SD 1.14) ($p<0.001$) (**Table 4**). Similarly, participants who viewed the health warning messages had stronger intentions to reduce red meat consumption in the next seven days (mean 2.45, SD 1.22) compared to those who viewed the environmental warning messages (mean 2.19, SD 1.17) ($p<0.001$).

### Variation of health and environmental messages (within-subjects)

Specific health harms elicited PME scores ranging from 2.46 to 2.75 (**Table 5**). After adjusting for multiple comparisons, nine statistically significant differences ($p<0.05$) between specific harms emerged. Early death and colon and rectal cancer messages elicited higher PME ratings than cardiovascular disease ($p = 0.008$ and $p = 0.001$, respectively). The diabetes message elicited lower PME ratings than all other messages ($p<0.001$ for all 7 comparisons).

A narrower range of PME was observed across different environmental messages (range, 2.20 to 2.32). After adjusting for multiple comparisons, there were three statistically significant differences between environmental harms ($p<0.05$). Environment performed better than carbon footprint ($p = 0.001$) and water shortages ($p = 0.032$). Water pollution performed better than carbon footprint ($p = 0.036$).

### Most discouraging health and environmental messages

For health messages, early death was most frequently selected as the most discouraging message (**Fig 3**, Panel A; p-values presented in **S2 Table**). Early death was selected significantly more often than all other health messages. For environmental messages, climate change, which leads to extreme weather events, was selected the most frequently, but there were not

**Table 4. Perceived message effectiveness (PME)\* and intention to reduce meat consumption in the next seven days\* between health and environmental messages.**

| Measure | Health Messages | Environmental Messages | p-value |
|---|---|---|---|
| | Mean (SD) | Mean (SD) | |
| **PME** | 2.66 (1.18) | 2.26 (1.14) | $<0.001$ |
| **Intention to reduce meat consumption in the next 7 days** | 2.45 (1.22) | 2.19 (1.17) | $<0.001$ |

\*Five-point Likert scale ranging from 1 "Not at all" to 5 "A great deal".

†Results from t-test between health and environmental messages.

**Table 5. Perceived message effectiveness (PME)\* of health and environmental harms, in descending order of PME, from an online experiment among 1,199 U.S. adults (March 2020).**

| Warning Message | Mean† | Standard Deviation |
|---|---|---|
| **Health Harm** | | |
| Colon and rectal cancer | 2.75[A] | 1.28 |
| Early death | 2.73[A] | 1.34 |
| Several types of cancer | 2.70[AB] | 1.30 |
| Stroke | 2.69[AB] | 1.24 |
| Colon cancer | 2.67[AB] | 1.26 |
| Heart damage | 2.66[AB] | 1.24 |
| Cardiovascular disease | 2.63[B] | 1.20 |
| Type 2 Diabetes | 2.46 | 1.21 |
| **Environmental Harm** | | |
| Environment | 2.32[A] | 1.24 |
| Water pollution | 2.30[AB] | 1.24 |
| Planet | 2.29[ABC] | 1.29 |
| Climate change, which leads to extreme weather events | 2.27[ABC] | 1.23 |
| Deforestation | 2.27[ABC] | 1.25 |
| Global warming | 2.26[ABC] | 1.24 |
| Greenhouse gas emissions | 2.24[ABC] | 1.20 |
| Climate change | 2.23[ABC] | 1.23 |
| Water shortages | 2.22[BC] | 1.27 |
| Carbon footprint | 2.20[C] | 1.17 |

\*Five-point Likert scale ranging from 1 "Not at all" to 5 "A great deal."

†Means sharing a letter in the superscript are not significantly different from one another at the 5% level.

‡Results from mixed-effects linear regression, with separate models for health messages and for environmental messages.

statistically significant differences between messages in likelihood of being selected as most discouraging (**Fig 3, Panel B**).

## Predictors of PME

The results from the OLS regressions on personal characteristics and PME are presented in **Table 6**. For both health and environmental harms, messages elicited higher PME among younger adults (age 18–25) compared to older adults (age 45 and older). Messages also elicited higher PME among participants who had reduced their red meat consumption over the past year compared to those who had not reduced their consumption ($b$ = 0.57 and 0.68 for health and environment, respectively, $p < .001$).

Health messages elicited higher PME scores among women than men ($b$ = 0.22, $p < 0.05$). This relationship was not statistically significant for environmental messages. Environmental messages elicited higher PME scores among Hispanic/Latinx participants compared to Not Hispanic/Latinx participants ($p$ = 0.01); among liberal participants compared to moderate ($p$ = 0.009) and conservative ($p$ = 0.001) participants; and among participants with an annual household income of $100,000 or more compared with participants with an annual household income of $24,999 or below ($p$ = 0.002). Health messages elicited higher PME scores among Black participants compared to white participants ($p < .05$); and among participants with a higher annual household income compared to $24,999 or below, except for $75,000-$99,999.

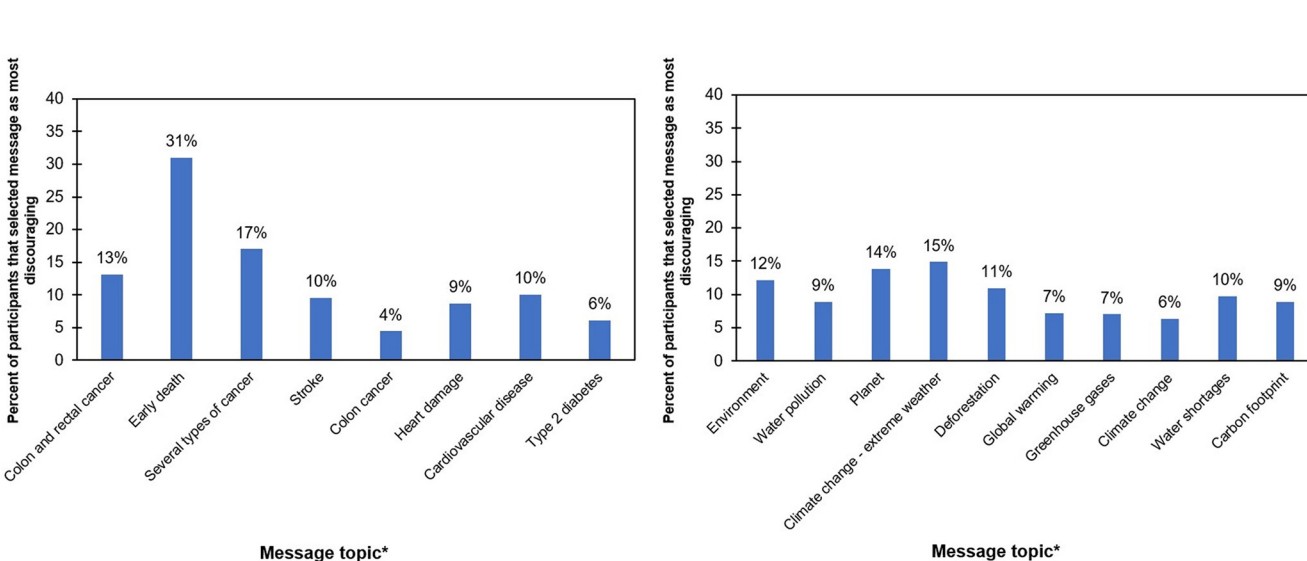

**Fig 3.** Message selected as most discouraging by U.S. adults randomized to the health (Panel A, n = 587) or environmental (Panel B, n = 584) group (March 2020). Panel A: Health, Panel B: Environment. *Message topic is presented rather than full message text for brevity. See Table 2 for complete message text. Message topics are arranged from highest to lowest PME score.

### Behavioral framing and causal variant tasks

Perceived discouragement ratings for different causal variants ranged from 2.41 to 2.62. There were no differences in perceived discouragement between causal variants after correcting for multiple comparisons. For the behavioral framing experiment, discouragement ratings ranged from 2.32 to 2.59. As with the causal variant experiment, differences were not found to be statistically significant in this exploratory study.

## Discussion

To our knowledge, this is one of the first studies to develop and compare health and environmental warning messages about red meat. In our sample of US adults, we found that red meat consumers perceived health warning messages to be more effective than environmental warning messages. Moreover, across specific health warning messages, those for colon and rectal cancer and early death were perceived to be most effective. Red meat warnings for both health and the environment elicited higher perceived message effectiveness ratings among younger adults and those reporting that they had reduced their red meat consumption over the past year. The relationship between other personal characteristics and PME varied depending on warning topic (health vs. environmental). PME from health warnings was higher among Black participants compared to white participants. For environmental warnings, PME was higher among Hispanic/Latinx participants, participants with an annual household income of $100,000 or more, and liberal participants.

Warnings about red meat could help inform consumers about the health and environmental impacts of these products [63]. Current awareness of the adverse health impacts likely is low given the lack of public health campaigns focused on red meat's health harms. Further, red meat is highly marketed. For example, the Beef Industry Council-funded ad campaign that promotes the consumption of red meat ("Beef. It's what's for dinner."), administered by the U. S. Department of Agriculture, is recognized by 88% of American adults [64]. Public confusion

**Table 6. Results of linear regression of demographic and behavioral characteristics and perceived message effectiveness (PME) of health (n = 587) and environmental messages (n = 587).**

| Characteristic | Health Messages (n = 587) | | Environmental Messages (n = 587) | |
|---|---|---|---|---|
| | Coefficient (SE) | p-value | Coefficient (SE) | p-value |
| **Age** | | | | |
| 18–25 | Referent | Referent | Referent | Referent |
| 26–34 | -0.26 (0.16) | 0.10 | -0.02 (0.13) | 0.87 |
| 35–44 | -0.10 (0.16) | 0.55 | -0.08 (0.14) | 0.57 |
| 45–54 | -0.32 (0.17) | 0.05 | -0.35 (0.14) | 0.01 |
| 55–64 | -0.35 (0.17) | 0.04 | -0.45 (0.14) | 0.001 |
| 65 or older | -0.63 (0.17) | 0.00 | -0.45 (0.14) | 0.001 |
| **Gender** | | | | |
| Male | Referent | Referent | Referent | Referent |
| Female | 0.22 (0.09) | 0.02 | 0.11 (0.08) | 0.16 |
| **Race** | | | | |
| White | Referent | | Referent | Referent |
| Black | 0.35 (0.16) | 0.03 | 0.01 (0.13) | 0.92 |
| Other race | 0.19 (0.16) | 0.23 | 0.17 (0.14) | 0.21 |
| **Ethnicity** | | | | |
| Not Hispanic/Latinx | Referent | Referent | Referent | Referent |
| Hispanic/Latinx | 0.13 (0.15) | 0.40 | 0.37 (0.14) | 0.01 |
| **Education** | | | | |
| Less than high school or high school diploma | Referent | Referent | Referent | Referent |
| Associate or technical degree | -0.10 (0.12) | 0.41 | -0.07 (0.12) | 0.51 |
| 4-year college degree | 0.11 (0.13) | 0.36 | 0.02 (0.10) | 0.86 |
| Master's degree or graduate degree | -0.02 (0.16) | 0.91 | 0.16 (0.14) | 0.26 |
| **Annual Household Income** | | | | |
| $24,999 or below | Referent | Referent | Referent | Referent |
| $25,000-$49,999 | 0.31 (0.02) | 0.02 | 0.20 (0.11) | 0.08 |
| $50,000-$74,999 | 0.35 (0.14) | 0.01 | 0.16 (0.12) | 0.19 |
| $75,000-$99,999 | 0.07 (0.17) | 0.68 | 0.20 (0.14) | 0.15 |
| $100,000 or more | 0.38 (0.16) | 0.02 | 0.42 (0.13) | 0.002 |
| **Political Affiliation** | | | | |
| Liberal | Referent | Referent | Referent | Referent |
| Moderate | 0.02 (0.11) | 0.88 | -0.26 (0.10) | 0.009 |
| Conservative | -0.22 (0.12) | 0.08 | -0.40 (0.12) | 0.001 |
| **Frequency of Red Meat Consumption in Past 30 Days** | | | | |
| Less than once per week | Referent | Referent | Referent | Referent |
| 1 time per week | -0.02 (0.20) | 0.92 | -0.49 (0.17) | 0.004 |
| 2–3 times per week | -0.20 (0.18) | 0.26 | -0.53 (0.14) | 0.00 |
| 4–6 times per week | -0.39 (0.20) | 0.05 | -0.72 (0.16) | 0.00 |
| Once a day or more | -0.34 (0.22) | 0.12 | -0.69 (0.18) | 0.00 |
| **Change in Red Meat Consumption in Past Year** | | | | |
| Did not reduce | Referent | Referent | Referent | Referent |
| Reduced | 0.57 (0.10) | 0.00 | 0.68 (0.09) | 0.00 |
| **Believe that climate change is caused by human activities** | | | | |
| Strongly or somewhat disagree* | - | - | Referent | Referent |
| Neither agree nor disagree | - | - | 0.30 (0.15) | 0.05 |
| Somewhat agree | - | - | 0.54 (0.12) | 0.00 |

*(Continued)*

**Table 6.** (Continued)

| Characteristic | Health Messages (n = 587) | | Environmental Messages (n = 587) | |
|---|---|---|---|---|
| | Coefficient (SE) | p-value | Coefficient (SE) | p-value |
| Strongly agree | - | - | 0.73 (0.12) | 0.00 |

*Strongly and somewhat disagree categories were collapsed into one category after initial regression post-hoc tests showed no significant difference in PME between these groups.

†Results from multivariate linear regression with OLS, with separate models for health messages and for environmental messages.

over the health effects of red meat has been exacerbated by inconsistent scientific messaging. Most recently, for example, the *Annals of Internal Medicine*, a journal published by the American College of Physicians, published a review article that concluded that there was insufficient evidence to support that red or processed meat consumption was associated with poor health outcomes [65]. This article received a widespread critique by nutrition scientists due to flaws in study design and interpretation bias [66]. Despite this critique, the article garnered widespread media coverage in outlets such as the *New York Times* [67]. In the same year, the New York Times also covered the EAT-*Lancet* report, which includes recommendations to limit red meat intake, in multiple articles [68,69]. The rapid and contradictory nature of reporting around red meat may contribute to consumer confusion.

In this study, health warnings elicited higher ratings of PME than did environmental warnings. The reasons for these differences are unclear. One possible explanation is that environmental warnings elicit stronger feelings of defensiveness among consumers [70] than health warnings. For example, abstinence from meat can be viewed as threatening [71]. If consumers felt more judged by the environmental messages, perhaps this led to lower ratings of PME. Similarly, research on graphic tobacco warnings has found that some messages elicit reactance, or cognitive or emotional resistance to a message, which suppresses warnings' effects on intentions to quit smoking [72]. More research will be needed to understand the degree to which environmental warning messages elicit reactance and potentially blunt the impact of the warnings.

Another possibility is that baseline awareness about the environmental damage caused by red meat is low, leading to lower PME. According to a recent report by the Yale Program on Climate Change Communication, only 27% of Americans think that beef contributes to global warming "a lot" and 30% think it contributes "a little" or "some" [73]. Further, the same survey found that 91% of consumers cited health as at least moderately important as a reason why they consumed plant-based foods, compared to 64% that cited reducing global warming as at least moderately important [73]. This is consistent with a previous nationally representative survey that found people who are reducing meat consumption more commonly cite health than environmental reasons, and the relationship we observed in this study is consistent with those findings [74]. Warnings on red meat may help increase public awareness and perceptions of risk of these environmental harms. In turn, higher public awareness about the environmental damages caused by red meat may increase the effectiveness of the messages. More research will be needed to understand consumers' reactions to environmental messages about red meat, particularly as public attitudes and knowledge shift over time.

Additionally, the difference in PME ratings between the health and environmental messages could be due to differences in the types of causal variants used. All of the health warning messages included the phrase "increases your risk of," while the environmental warnings used "contributes to," "increases," and "harms." On the other hand, the results from our causal

variant task suggested that the type of causal language used does not meaningfully influence PME for messages about red meat. We tested three causal variants linking red meat with one health harm, obesity ("Eating red meat is associated with, increases your risk, or increases your chances of obesity") and found that there were no differences in PME by causal variant. However, we did not test different causal variants for other health harms, nor for any environmental harms, and thus, it would be useful for future research on messages about red meat to explore whether different causal variants are perceived as more or less effective.

Within the messages focused on health harms, there were few differences in PME between different health topics. The exception to this pattern was type 2 diabetes, elicited lower PME ratings than all other health harms tested in this study. These differences may be because other outcomes, such as cancer and mortality, may be perceived as more damaging to health than type 2 diabetes. Furthermore, these differences could be due to the fact that the general public has overall low awareness about the association between red meat intake and type 2 diabetes [63].

Similar to the results for health harms, there were few differences in PME across specific environmental harms, with little variability between the top rated and lowest rated messages. The exception to this pattern is that the environment message elicited higher PME than the carbon footprint, water shortage, and climate change messages. These results are consistent with a recently published study by Wistar et al. which also found that a general environment message was perceived as more effective at discouraging red meat consumption than other environmental topics [75]. In contrast, it was surprising that the carbon footprint message elicited lower perceived message effectiveness as many existing environmental warning labels focus on carbon footprints. Lower PME for the carbon footprint message could reflect that many consumers do not understand the meaning of carbon footprints [76,77]. Our findings that many health and environmental harms elicited similar PME highlights that practitioners and policymakers have a wide variety of options for which harms to include in warnings.

We assessed the concept of PME two different ways in this study. Although PME rating of individual warnings was the primary outcome [35], we also asked participants to select which message of all the messages would most discourage them from consuming red meat because forcing participants to select one message might amplify subtle differences in reactions to each warning. For health messages, PME ratings were highest for messages about colon and rectal cancer, followed by early death, and several types of cancer, which all had very similar PME ratings. The overall pattern of top messages was the same when participants were forced to select the most discouraging message compared to when they rated each message individually, though the differences between messages appeared to be amplified in the former. For example, a substantial margin separated the proportion of participants who selected early death as the most discouraging harm (31% of participants) compared to several types of cancer or colon and rectal cancer (selected by 17% and 13% of participants, respectively), despite these harms performing similarly in PME ratings. These differences could reflect differences in measurement approaches (rating vs. forced choice), or that the 3-item PME scale measures two constructs in addition to discouragement (concern and unpleasantness). Overall, results from both sets of questions suggest that messages focused on cancer and especially early death may be particularly promising for eliciting behavior change.

For the environmental messages, results were largely consistent regardless of whether participants were rating messages on PME or selecting the most discouraging message. Specifically, there was little variability in both PME ratings (range of mean PME scores, 2.20 to 2.32) and in the proportion of participants who selected each message as the most discouraging (range of %s, 6% to 15%). This pattern of results suggests that a variety of environment-related messages would perform similarly.

With regards to demographic and personal predictors of PME, health warning messages elicited a higher PME among Black compared to white participants. It is possible this pattern was seen because racial inequities persist in chronic diseases such as type 2 diabetes and cardiovascular disease, wherein the prevalence is higher among Black as compared to white adults [78]. This higher message effectiveness may also be because at the population level Black individuals are already reducing their meat consumption more than white individuals [79], and so may be more receptive to these types of warning messages. If this pattern of results holds in real-world settings, health warning messages on red meat could serve as a tool for reducing racial inequity in chronic diseases.

Political affiliation and belief in human-caused climate change were the strongest predictors of PME in the environmental message model. As we hypothesized, stronger beliefs in human-caused climate change were associated with higher perceived effectiveness of the environmental messages. This finding is consistent with the findings from "Climate Change and the American Diet" [73]. While these findings indicate that those who do not believe that human-caused climate change may be harder to reach with environmental messaging, only 18% of our sample strongly or somewhat *disagreed* with the statement that human-caused climate change is occurring. This finding is consistent with other national polls of belief in climate change that show that the majority of US residents believe in human-caused climate change [80].

Environmental messages elicited higher PME from liberals than from moderates and conservatives. This finding is consistent with climate change literature that has examined concern for climate change based on political belief [81,82]. This relationship was not significant in the analysis of health messages. Thus, health messages may be perceived as less politically controversial in the US.

In both our health and environmental analyses, messages were less effective among older adults compared to 18-25-year-olds (45 and older for environment warning messages; 55 and older for health warning messages). Young adults are the fastest growing age group for vegetarians and vegans; 12% of U.S. adults aged 18–29 report eating mostly vegetarian or vegan, compared to 5% of U.S. adults aged 50 and older [79]. Though our survey contained only red meat consumers, younger people may be more willing to reduce their meat consumption, perhaps contributing to their stronger perceptions that warning messages would be effective at reducing red meat consumption.

For both health and environmental messages, participants who had not reduced their red meat consumption over the past year rated messages as less effective than participants who had reduced their red meat consumption. Higher self-reported consumption of red meat was significantly associated with lower PME ratings for environmental messages. Again, this relationship between lower red meat consumption and higher PME for environmental messages was consistent with the recent study published by Wistar et al. [75]. No such relationship between red meat intake and PME emerged for health messages.

All causal variants and behavioral frames performed similarly. These findings suggest messages can take a variety of forms and perform well. These findings are important because subtle differences in warning language could affect warnings' legal viability in the US; if a variety of warnings are similarly effective, policymakers can pursue those most likely to clear potential legal challenges.

This research was conducted with the goal of informing future policies to inform consumers about the harms of red meat and discourage consumption. Although warning labels on other food products, such as sugary beverages, have been successful in several countries, policies focused on red meat have not been implemented, so legal and political feasibility in the US must be considered. In the US, warning may face First Amendment challenges [83]. Furthermore, the strong political power of the meat industry [83,84], as well as deep-rooted ideologies

of carnism and neoliberalism [84], could make implementing meat warnings politically challenging.

Although perceived effectiveness is a marker for messages' potential to change behavior, more research is needed to assess whether red meat warnings influence red meat purchasing and consumption behaviors. To our knowledge, only one study has investigated the impact of health and environmental warnings on outcomes related to red meat purchases [27]. That study found that participants perceived health warnings and combined environmental and health warnings as more effective than environmental warnings. However, the warnings did not affect participants' selections in a product choice task. Given that previous research has shown that perceived message effectiveness is predictive of behavioral change [35], it is unclear why even the health warnings did not reduce selection of red meat in that study. One possibility is that the behavioral task may not have been properly powered to detect differences [27], or that the simplistic nature of an online choice experiment does not reflect how warnings would influence purchasing decisions in the real world. Future research would be useful to understand whether such warnings would discourage meat purchases in a more realistic food retail environment.

Strengths of this study include the large sample of US residents and use of a randomized experiment. Limitations includes the use of a convenience sample, though the Cloud Research sample has been shown to more closely reflect the US population on sociodemographic characteristics than other online samples [56]. Furthermore, this was an exploratory study without an *a priori* power calculation, and it is possible that there were differences in consumer responses to messages that we were not powered to detect. Additional limitations include that this study did not assess behavioral outcomes, such as purchases of red meat, although previous research on communication campaigns has found that PME, our primary outcome, is predictive of behavioral change [85]. Further research will clarify how consumers alter food purchases in response to red meat warnings. Another limitation is that data were collected in late March 2020, while coronavirus was spreading across the US. The spread of coronavirus may have elicited higher PME in health warning messages, particularly early death, given heightened concern about health harms during this time period. Public health agencies widely communicated the elevated risk of severe COVID-19 infection and death for individuals with pre-existing conditions [86]. Many of these pre-existing conditions are associated with diet, such as type 2 diabetes and cardiovascular disease [87]. Additionally, research has found that the COVID-19 pandemic increased mortality salience, leading to increased health behavior intentions [88,89]. Future research should examine how the salience of health and mortality risks caused by the coronavirus pandemic may impact reactions to health warning messages.

## Conclusions

Reducing red meat consumption in the US is critical for human and planetary health. This study developed several evidence-based health and environmental warning messages for red meat. Results of this exploratory study suggest that health warning messages are perceived to be more effective than environmental messages. Nonetheless, the inclusion of environmental waring messages is an opportunity to educate consumers about the environmental harms of consuming red meat. Future work will clarify the impact of these messages on meat purchases and consumption.

## Supporting information

**S1 Appendix. Codebook.**
(DOCX)

**S1 Table. Adjusted perceived message effectiveness (PME)** * **and intention to reduce meat consumption in the next seven days** * **between Health and Environmental Messages.** *Five-point Likert scale ranging from 1 "Not at all" to 5 "A great deal" †Results from linear regression, adjusting for meat consumption as compared to a year ago.
(DOCX)

**S2 Table. P-Values from Z-Tests comparing messages selected as most discouraging by U. S. adults randomized to the health (n = 587) or environmental (n = 584) group.**
(DOCX)

## Acknowledgments

The authors thank Christina Chauvenet for project management, statistical analysis, and manuscript drafting assistance. The authors thank Hannah Rayala for the design of the warning images. The authors also thank Isabella Higgins and Sarah Frank for feedback on survey design and methodology.

## Author Contributions

**Conceptualization:** Lindsey Smith Taillie, Marissa G. Hall, Anna H. Grummon, Lindsay M. Jaacks.

**Writing – original draft:** Lindsey Smith Taillie, Lindsay M. Jaacks.

**Writing – review & editing:** Lindsey Smith Taillie, Carmen E. Prestemon, Marissa G. Hall, Anna H. Grummon, Annamaria Vesely, Lindsay M. Jaacks.

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
