## [Decision Letter · Decision Letter 0]

3 Jun 2021

PONE-D-21-09001

Developing health and environmental warning messages about red meat: An online experiment

PLOS ONE

Dear Dr. Taillie,

Thank you for submitting your manuscript to PLOS ONE. After careful consideration, we feel that it has merit but does not fully meet PLOS ONE’s publication criteria as it currently stands. Therefore, we invite you to submit a revised version of the manuscript that addresses the points raised during the review process.

We look forward to receiving your revised manuscript.

Kind regards,

John W. Apolzan, PhD

Academic Editor

PLOS ONE

Additional Editor Comments:

This is more of a reject with opportunity to resubmit than a major revision. Revision doesn't guarantee acceptance and both reviewers had concerns. Please thoroughly respond to all reviewer comments. Please pay close attention to R1's statistical analyses concerns. Please also provide power calculations for the trial and if primary are powered please further clarify for secondary within entire manuscript (i.e. exploratory). If not prospectively powered, clarify in abstract and throughout manuscript pilot study or exploratory. With R2, please perform a sensitivity analyses pre vs. COVID-19 to see if it affects results.

Journal Requirements:

Reviewers' comments:

Reviewer's Responses to Questions

**Comments to the Author**

1. Is the manuscript technically sound, and do the data support the conclusions?

Reviewer #1: Partly

Reviewer #2: Partly

2. Has the statistical analysis been performed appropriately and rigorously? 

Reviewer #1: No

Reviewer #2: Yes

3. Have the authors made all data underlying the findings in their manuscript fully available?

Reviewer #1: Yes

Reviewer #2: No

4. Is the manuscript presented in an intelligible fashion and written in standard English?

Reviewer #1: No

Reviewer #2: Yes

5. Review Comments to the Author

Reviewer #1: Comments to authors

This study investigates the effectiveness of health and environmental warning messages on consumers’ intention to reduce red meat consumption via an online experiment. The authors claim that health warning is more effective than environmental messages. The topic is timely and relevant. However, the measurement of the effectiveness is poorly designed. Based on the results resented by far, I am far from convinced that the health warning messages are more effective.

The most important concern that I have for this study is the choice of the effectiveness measurement. In the current version, the authors use perceived message effectiveness (PME) as the key measurement. PME is a self-report Likert scale about the message, e.g., I think this message discourage me from wanting to eat red meat a great deal or not at all.

If we assume that the message works via the following mechanism: (a)messages  (b) change thoughts  (c) change behavior (e.g., consumption frequency). What the authors measured is only the first step (from a to b) - the impacts of message on thoughts not behavior. Given that there are field studies already successfully measured the impacts of nutrition education on direct consumption outcomes, I find that a study only investigating the partial impacts of the message on people mind could provide limited solid and novel insights to the existing literature.

Further, a large amount literature has demonstrated that the hypothetical bias associated with self-report behavior intention, and such bias can be particularly large when social desirability bias exists. I believe both health and environmental message could induce social desirability bias which could exacerbate hypothetical bias.

In term of statistical analysis, I strongly disagree that authors directly calculated means of Liker scale, e.g., 5= a great deal and 1= not at all, and the average is 2.6. By taking average of those scales, the authors are assuming the distance between 1=not at all and 2 = somewhat not important are equal to the distance between 4=somewhat important and 5=a great deal, which is a very poor assumption and is also not the common practice. Corresponding to that, instead of using linear regression model (table 5), the author should use ordered logit model.

Also, I strongly disagree with the author that, because messages are randomly assigned, the balance tests are therefore not required. Balance tests are essential to demonstrate that your randomization was successful. Unsuccessful randomization is fairly common even when researchers thought they randomly assigned subjects to treatment groups.

For the experiment design, shouldn’t there be a baseline group where they received a placebo information so that their attitude can be used as the gold standard benchmark?

Minor comments

The presentation of the table requires a lot of work. The presentation of Table 1 is horrible. This single table takes 8 pages with a lot of blank areas on many pages. For table 3, the author can easily add one more column to compare the differences between Health Warning and Environmental Warnings groups as the balance test. As for table 5, there are more professional presentation to report regression results.

Reviewer #2: This study explored participants’ reactions to health and environmental warnings messages about red meat. The manuscript is well written and easy to follow. I have included general and specific notes below on areas of the paper that I believe could be strengthened:

GENERAL

You mention at the very end of your limitations section that COVID may have elicited higher PME in health warning messages, especially for early death, which seems likely given its salience during the study period. This seems to be a very large limitation of this work, and should be discussed in more detail. Did you collect any data before the nationwide quarantine (i.e., early March)? Is there any way you can do a sensitivity analysis looking at responses from earlier in the pandemic (March) compared to later (April) to see if the effectiveness of the warnings changed over time? In the discussion, I think it would be valid to spend more time reflecting on COVID’s influence on this study. Do you think health warnings will continue to be salient, given the ongoing pandemic? Is there any scientific evidence that you can cite that supports or negates the idea that the salience of health risks created by COVID may have influenced your findings?

I think this manuscript could benefit from a more in-depth discussion of the application of red meat warnings. How policy-relevant is this work? Is there any way such warnings could be legally required, either in the US or internationally? Do you foresee these warnings being used in educational materials or PSAs?

INTRODUCTION

p. 3, line 51: Can you give the reader context for what 284 grams of red meat looks like in a person’s diet, perhaps in servings of steak or burgers?

p. 3, line 60: typo: “knowledge of a safe consumption levels”

METHODS

General: Could you standardize your language throughout the methods section to have consistent terminology for your main experiment, the behavioral framing task, and the causal variant task? I found it a little confusing when you referred to your main experiment as the “between subjects” experiment, because it appears that all three of the tasks in your study were between subjects (in the second and third tasks, participants were randomized to see only one message, correct?).

p. 14, line 122: Can you explain why you tested 8 health messages and 10 environmental messages, instead of 8 and 8? Even just a few additional words at the end of the sentence in lines 122-123 would be helpful. Something like, “we created a final list of eight health messages…to be used in this experiment CHOSEN BASED ON X CRITERIA”

p. 14, lines 123-124: You list 9 health harms here (not 8), and mention colon and rectal cancer twice—could you clarify?

p. 14, line 126: Can you separate your list with semicolons to make it clear that you didn’t accidentally list climate change twice? I had to reread that sentence to understand what you were trying to say.

p. 17, line 188: Can you be more specific about your primary outcome, given that you’re assessing PME in three separate tasks? ie, add a note that it’s PME for the health/environment warning messages in the main experiment

p. 19, lines 218-222: While balance tests may be unnecessary for randomized experiments, I still think it’s useful to check for differences in demographics and if one is found, conduct a sensitivity analysis on your primary outcome to ensure it isn’t affecting your results.

p. 19, line 223: “between-subjects’ arms”: see general comment under methods section heading. Could you refer to this as your main experiment (or something like that)?

p. 19, line 228: “most discouraging message”: You had previously mentioned asking participants which message was most discouraging under Procedures lines 162-163, but I didn’t see that as an outcome of interest in the measures section.

p. 19, lines 230-236: How did you determine which predictors to include in your models?

RESULTS

Table 3: When you refer to “meat” in this table, is that red meat? If so, can you modify to “frequency of red meat consumption” and “red meat consumption as compared to a year ago”?

pp. 24-25, lines 261-269: Can you list the p-values for the significant findings you’re reporting, beyond just saying they’re <0.05? It’s hard to understand what a significant difference of 0.1 on the PME scale means without added context.

Table 4: Why was the SD for global warming so small (0.05) compared to the others (~1.2-1.3)?

Figure 3: How was the order of messages on the X axis determined? Could you order them from highest to lowest PME score (and add a note at the bottom of the figure that they’re arranged that way) so that readers can see the relationship between the discouragement outcome vs. PME outcome, given that they’re both trying to predict behavior?

p. 29, lines 290-293: There appear to be other significant differences in predictors for PME for the health and environmental message models individually that weren’t commented on in the text (e.g., income, political affiliation, ethnicity)—could you either add those results to the text or provide justification for excluding them?

DISCUSSION

General: Your main takeaway for differences between warning messages was that colon and rectal cancer and early death were most effective (based on PME), but based on your discouragement outcome, only early death was “effective”. I’m curious as to why you chose to highlight both outcomes (PME and discouragement) in the results—which do you think is most predictive of actual behavior change? What do we gain by using the other outcome?

General: I think it’s worth commenting on the varied phrasing of the messages you tested, given that your health warnings all used “increases your risk of” but the environmental warnings never used the word “risk” and instead used words like “contributes to”, “increases”, and “harms.” Is there reason to believe that the word “risk” contributed to the health warning messages’ success? Could you craft an environmental message that frames the warning as health risks to the environment? This seems worthy of mention and future research, given your findings.

p. 30, lines 307-308: I think it’s worth spending a little more time talking about significant predictors of PME for health and environment warnings, especially given ethnic and income differences that could be of use for future messaging research.

p. 30, lines 309-310: Does your study provide evidence that warnings about red meat could help inform consumers about the health and environmental impacts of products? If so, can you explicitly say that, and if not, can you temper this statement?

p. 30, lines 349-355: Couldn’t it also be possible that type 2 diabetes had the lowest PME because most people associate it with sugar and carbs, not red meat? You also found differences in the environment messages that seem worth commenting on—why do you think the environment message performed so much better than carbon footprint? This seems useful to mention given that many labels focus on carbon footprints.

p. 32, lines 356-374: Can you also speak to the other significant differences in predictors that you observed in Table 3? You didn’t mention some of these results until now, so I recommend adding them to the results section before they’re mentioned in the discussion.

p. 34, lines 394-398: You didn’t see statistically significant differences in the causal language or eating behavior language experiments, but could that be because you weren’t powered to see significant effects, given that your sample sizes were much smaller per group for those analyses? Can you provide an explanation of what your study was powered to find? If you were underpowered to detect differences in effects for these two experiments, I would recommend tempering your language here and including this as a limitation.

CONCLUSIONS

I don’t think your study found that health and environmental harms could be used in warning messages about red meat—your study created warning messages about red meat that included health and environmental harms, and you found that health warnings appeared to be more effective.

6. PLOS authors have the option to publish the peer review history of their article (what does this mean?). If published, this will include your full peer review and any attached files.

Reviewer #1: No

Reviewer #2: No

---

## [Author Response · Author response to Decision Letter 0]

13 Jan 2022

Dr. Emily Chenette

Editor-in-Chief

PLOS One

Dear Editor,

We thank you and the reviewers for your comments on our recently submitted manuscript titled “Developing health and environmental warning messages about red meat: An online experiment” (ID PONE-D-21-09001). We appreciate the thoughtful comments from the reviewers and believe the revised manuscript is significantly stronger for having incorporated their suggestions. 

We are attaching a revised version of the manuscript with track changes. We are also attaching a table to this submission that contains point-by-point responses to the reviewers’ comments.

Another change is that co-author Christina Chauvenet has requested to be moved from the author list to the acknowledgements section as due to a job change, she can no longer to contribute to reviewing or contributing to the manuscript. We have also added Carmen Prestemon as a co-author. Carmen is a research assistant who contributed to the project during the initial phases and also contributed writing, editing, and analysis during the revision; we believe that her contributions warrant authorship. 

We look forward to hearing from you regarding our submission. We would be happy to respond to any further questions and comments that you might have.

Kind regards,

Lindsey Smith Taillie, PhD

Assistant Professor, Department of Nutrition 

University of North Carolina at Chapel Hill

---

## [Decision Letter · Decision Letter 1]

16 Feb 2022

PONE-D-21-09001R1Developing health and environmental warning messages about red meat: An online experimentPLOS ONE

Dear Dr. Taillie,

Thank you for submitting your manuscript to PLOS ONE. After careful consideration, we feel that it has merit but does not fully meet PLOS ONE’s publication criteria as it currently stands. Therefore, we invite you to submit a revised version of the manuscript that addresses the points raised during the review process.

We look forward to receiving your revised manuscript.

Kind regards,

John W. Apolzan, PhD

Academic Editor

PLOS ONE

Journal Requirements:

Reviewers' comments:

Reviewer's Responses to Questions

**Comments to the Author**

1. If the authors have adequately addressed your comments raised in a previous round of review and you feel that this manuscript is now acceptable for publication, you may indicate that here to bypass the “Comments to the Author” section, enter your conflict of interest statement in the “Confidential to Editor” section, and submit your "Accept" recommendation.

Reviewer #2: (No Response)

2. Is the manuscript technically sound, and do the data support the conclusions?

Reviewer #2: Yes

3. Has the statistical analysis been performed appropriately and rigorously? 

Reviewer #2: Yes

4. Have the authors made all data underlying the findings in their manuscript fully available?

Reviewer #2: Yes

5. Is the manuscript presented in an intelligible fashion and written in standard English?

Reviewer #2: Yes

6. Review Comments to the Author

Reviewer #2: Thank you for your thoughtful edits and responses. A few more thoughts:

Lines 82-83, 96-97: You say that studies have not yet examined health warnings on red meat or developed warning messages about the environmental impact of red meat, but didn’t the study below do that? I understand that this study under review informed the behavioral study cited below and that study was conducted after this one, but it still was published first so I think you need to acknowledge it. You could change this language to something like, “before conducting this study, no research had examined health warnings…” etc.

Taillie, L.S., Chauvenet, C., Grummon, A.H. et al. Testing front-of-package warnings to discourage red meat consumption: a randomized experiment with US meat consumers. Int J Behav Nutr Phys Act 18, 114 (2021).

Lines 197, 405: you used “casual” instead of “causal”, just double check if that happened anywhere else in the paper.

Lines 255-256: You say that you corrected for multiple comparisons using Bonferroni’s method—can you specify whether the p-values presented in the paper are corrected, and if so, which ones?

Figure 3: I previously commented about changing this figure and although you said that you made the change, it does not appear to have changed in the revision you submitted. My original comment stands: I think it would be more useful to reorder the x axes in Panels A and B so that messages are ordered from highest to lowest PME score from left to right. You added a footnote saying that message topics are arranged from highest to lowest PME score, but they don’t appear to be.

Discussion: I know your discussion is already quite long, but I think the Taillie et al. 2021 paper cited above deserves some acknowledgement in the discussion, given that you are calling for that work to be done at the end of your paper but it already has been done and published. In that published paper you did not comment on how your findings compared to this paper (because it wasn’t published yet), so it seems important to note in this paper that although you found significant differences in this paper, you did not see those differences play out in an actual behavioral task. I know it was one single task and may not have been properly powered, but I think it is still worth discussing.

7. PLOS authors have the option to publish the peer review history of their article (what does this mean?). If published, this will include your full peer review and any attached files.

Reviewer #2: No

---

## [Author Response · Author response to Decision Letter 1]

25 Mar 2022

Thank you to the editor and reviewers for the opportunity. For specific responses to each comment, please see the response letter attached to this submission.

---

## [Editor Report · Decision Letter 2]

25 Apr 2022

Developing health and environmental warning messages about red meat: An online experiment

PONE-D-21-09001R2

Dear Dr. Taillie,

We’re pleased to inform you that your manuscript has been judged scientifically suitable for publication and will be formally accepted for publication once it meets all outstanding technical requirements.

Kind regards,

John W. Apolzan, PhD

Academic Editor

PLOS ONE
---

## [Editor Report · Acceptance letter]

15 Jun 2022

PONE-D-21-09001R2 

Developing health and environmental warning messages about red meat: An online experiment 

Dear Dr. Taillie:

I'm pleased to inform you that your manuscript has been deemed suitable for publication in PLOS ONE. Congratulations! Your manuscript is now with our production department. 

Kind regards, 

on behalf of

Dr. John W. Apolzan 

Academic Editor

PLOS ONE